# Active Films Based on Starch and Wheat Gluten (*Triticum vulgare*) for Shelf-Life Extension of Carrots

**DOI:** 10.3390/polym14235077

**Published:** 2022-11-23

**Authors:** Andrés Felipe Rivera Leiva, Joaquín Hernández-Fernández, Rodrigo Ortega Toro

**Affiliations:** 1Food Packaging and Shelf-Life Research Group (FP&SL), Food Engineering Department, Universidad de Cartagena, Cartagena de Indias 130015, Colombia; 2Chemistry Program, Department of Natural and Exact Sciences, San Pablo Campus, University of Cartagena, Cartagena 130015, Colombia; 3Department of Natural and Exact Sciences, Universidad de la Costa, Calle 58 # 55–66, Barranquilla 080002, Colombia; 4Chemical Engineering Program, School of Engineering, Universidad Tecnológica de Bolivar, Parque Industrial y Tecnológico Carlos Vélez Pombo km 1 Vía, Turbaco 130001, Colombia

**Keywords:** active films, starch, wheat gluten, cinnamon essential oil, turmeric essential oil, shelf life

## Abstract

The use of biodegradable biopolymers with the incorporation of active ingredients has been considered as an alternative to extend the useful life of food. Therefore, the objective of this research was to develop active films based on starch and wheat gluten, containing cinnamon and turmeric essential oils by using the solvent casting method. Different film formulations were made from wheat starch, gluten, glycerol, and essential oils of cinnamon and turmeric. The films were characterized according to their morphology, optical, thermal, antioxidant, and barrier properties. Subsequently, the active properties on baby carrots regarding weight loss, appearance, and fungal growth were evaluated. The results indicated that the starch-based films showed a slight decrease in moisture content with the addition of essential oils (up to 13.29%), but at the same time showed a significant reduction in water solubility (up to 28.4%). Gluten-based films did not present significant differences in these parameters, although the solubility in water tended to increase (up to 13.15%) with the addition of essential oils. In general, the films presented good thermal stability and antioxidant capacity, and in the carrot coating test, a decrease in weight loss of up to 44.44% and 43.33% was observed for the coatings based on starch and gluten with the addition of turmeric essential oil, respectively. Finally, films developed with cinnamon and turmeric essential oils are potential candidates for the design of biodegradable active packaging.

## 1. Introduction

Food products are exposed to external conditions throughout the production chain, i.e., harvesting, post-processing, distribution, transportation, storage, and delivery to the final consumer [1]. In the last decade, food packaging has been developing new biodegradable active and smart materials that extend shelf life, maintain quality, safety, and integrity, and avoid microorganisms’ proliferation and food oxidative reactions by incorporating active substances [2,3]. Along with protecting food products from environmental conditions and mechanical forces, active packaging plays an active role in quality and food preservation during the distribution process [3,4].

The use of biopolymers in packaging has increased considerably over the past few years due to their sustainable feedstock, biodegradability, and similar processing characteristics to existing thermoplastics [5]. Starch is one of the most researched and widely used raw materials in the production of biodegradable films due to its high bioavailability, low cost, biodegradability, and renewability [6]. Moreover, starch possesses good film-forming properties and chemical stability [7,8,9,10]. In addition, considering the scarcity of fossil fuel resources, starch is considered an alternative renewable agricultural resource with excellent film-forming ability, non-toxicity, and good gel stability. Furthermore, it can be made by standard techniques, such as solution casting and extrusion, and thus processed as a thermoplastic material using a plasticizer (urea, glycerol, sorbitol, glycerin) and water. However, starch-based films have poor resistance to water due to their hydrophilic nature. Therefore, blending starch with other polymers is a possible way of developing starch films [11,12].

On the other hand, wheat gluten has film-forming properties, and its high bioavailability and low cost makes it highly suitable for the preparation of biodegradable polymers [13]. Wheat gluten has good oxygen and carbon dioxide barrier properties in dry conditions, controlling cellulosic materials’ inherent gas permeability [14].

In order to confer active properties to biodegradable films, several substances can be incorporated into the biopolymers, such as essential oils (EOs) [15,16]. Many research studies have shown that the use of EOs inhibits the growth of a wide variety of Gram-negative bacteria, such as *Escherichia coli*, *Pseudomonas aeruginosa*, and *Salmonella typhimurium*, as well as Gram-positive bacteria including *Staphylococcus aureus*, *Listeria monocytogenes*, *Streptococcus pyogenes*, and *Alicyclobacillus acidoterrestris* [17,18]. Additionally, other reports have demonstrated the high antioxidant activity of EOs, which ranges from 70% to 95% inhibition of free radicals of 2,2-diphenyl-1-picrylhydrazyl (DPPH) and 2,2′-azino-bis-(-3-ethylbenzothiazoline-6-sulfonic acid) (ABTS), due to the presence of hydroxyl groups (–OH) in its chemical structure [13]. Due to the aforementioned antimicrobial properties and antioxidant capacity, EOs have been widely used to develop active packaging for food [19]. Among the most studied EOs are cinnamon and turmeric, which are composed of a wide range of volatile components such as terpenes, alcohols, acids, esters, epoxies, aldehydes, ketones, amines, and sulfides, among others, responsible for its strong biological activity [20,21,22,23]. The high antimicrobial and antioxidant properties of cinnamon EOs have been attributed mainly to cinnamaldehyde, cinnamate, cinnamic acid, trans-cinnamaldehyde, cinnamyl acetate, eugenol, L-borneol, camphor, caryophyllene oxide, b-caryophyllene, L-bornyl acetate, E-nerolidol, α-cubebene, α-terpineol, terpinolene, and α-thujene [24]. In the case of turmeric EOs, the high bioactivity has been attributed to curcuminoids that consists of curcumin and two related compounds: demethoxy curcumin and bisdemethoxycurcumin [25].

Since most EOs are volatile compounds, they require the use of manufacturing methods that are carried out at room temperature to preserve their bioactivity. In this sense, the most commonly used method for a laboratory biopolymer film formation is solvent casting that largely depends on polymer solubility [26]. This method consists of three basic steps: first, biopolymer solubilization in a solvent, which is chosen according to biopolymer chemical structure; second, the solution is poured into molds, which are usually Teflon-coated glass plates; and third, the casted solution is heat dried, wherein the solvent is evaporated to obtain a polymer film adhered to the molds with a homogeneous and continuous microstructure [27,28,29]. Lastly, the physical and chemical properties of the film are dependent on casting solution composition, wet casting thickness and drying conditions (e.g., temperature and relative humidity) [30].

In this context, the objective of this research was to develop active films based on starch and wheat gluten containing cinnamon and turmeric EOs by the solvent casting method. Initially, different formulations were prepared varying the EOs and biopolymers concentrations. Then, the active films were characterized in terms of their moisture content (MC), water solubility (WS), water absorption capacity (WAC), contact angle, morphology, color, thermogravimetric analysis (TGA), antioxidant capacity (DPPH IC_50_ and ABTS IC_50_) and water vapor permeability (WVP). Thereafter, the active films were applied on baby carrots to increase their shelf life and maintain their quality.

## 2. Materials and Methods

### 2.1. Materials

Wheat flour was purchased from the local city market, glycerol was provided by Panreac, and EOs were purchased from NowFoods.

### 2.2. Wheat Starch and Gluten Extraction

Starch extraction was carried out forming a suspension of wheat flour and distilled water, which was washed several times with distilled water and filtered with muslin cloth, which allowed the starch to pass through. The starch suspension was kept at rest in a container at a temperature of 5 °C for 12 h for starch decantation. After the decanting time, the aqueous phase was removed from the vessel and the starch was washed with distilled water. Once the starch-washing stage was completed, the starch was recovered for subsequent drying in the sun. Once it was completely dry, it was milled and sieved to reduce its particle size and make it uniform. The calculation of the starch extraction yield was made between the starch obtained and the total wheat used.

For gluten extraction, a dough was first formed from wheat flour and 2% NaCl, and the dough was kneaded until it had a consistency sticky to the touch. The dough was left to rest for 5 min, washed with water for the necessary time to remove the starch, and then the clean gluten was weighed and taken as a percentage of the initial dough. To obtain the dry gluten, the gluten obtained from was dried in an oven at 100 °C for 5 h to constant weight, then cooled and weighed.

### 2.3. Active Film Development

Starch films were obtained based on the solution casting method, relying on previous studies with minor modifications, making several formulations and varying the bioactive compounds [31]. The film-forming dispersion was prepared by placing the powdered starch in a 250 mL conical flask containing distilled water; the values used are described in Table 1. The dispersion was heated in a water bath at 90 °C for 30 min while stirring at 500 rpm for the starches to gelatinize completely. Subsequently, the heating was stopped, and the dispersion was allowed to cool to 40 °C. 25% glycerol was added to the starch and stirred at 150 rpm for 20 min. After cooling, 3% of the EO (Cinnamon essential oil or Turmeric essential oil), depending on the formulations, was added to the dispersion in relation to the weight of the starch. The film-forming dispersion was then poured onto a Teflon to obtain the wheat starch-based films [32].

Gluten films were obtained following the methodology of Olabarrieta et al. [33], with minor modifications, making different formulations as shown in Table 1, where the wheat gluten solution was prepared under stirring conditions by mixing wheat gluten powder and glycerol in ethanol. Deionized water was then added to the solution formed. The solution was heated at 75 °C for 20 min, then stirred for 10 min. After cooling, the EO (Cinnamon essential oil or Turmeric essential oil), depending on the formulations, was added. Finally, the solution was poured onto a Teflon for drying to obtain the wheat gluten-based films.

### 2.4. Film Characterization

#### 2.4.1. Thickness

Before testing, the thickness of all films was measured using a digital micrometer (S00014, Mitutoyo, Corp., Kawasaki, Japan) with ±0.001 mm accuracy. Measurements were performed and averaged at five different points, two in each end and one in the middle.

#### 2.4.2. Morphology

Cross-sections and surface sections of biodegradable films were studied using an optical microscope (Zeiss Primo Star HD cam, Jena, Germany) integrated into a high-definition camera with 40× magnification. The microphotographs were analyzed and processed through Image Pro-Plus version 5.1 computer program.

#### 2.4.3. Color

The optical parameters (luminosity (*L*∗), *a*∗, *b*∗, chrome (C∗ab), and hue angle (h∗ab)) of films were measured with a MINOLTA Spectro-colorimeter (Minolta Co., Tokyo, Japan). Total color differences (ΔΕ) with respect to the control film were also determined using Equation (1).
(1)$$\Delta E=\sqrt{{\left(\Delta a\;*\right)}^{2}+{\left(\Delta b\;*\right)}^{2}+{\left(\Delta L\;*\right)}^{2}}$$

#### 2.4.4. Moisture Content

The films were conditioned to a relative humidity of 53%. Moisture content was determined using a convection oven at 60 °C until a constant weight was obtained. Moisture was calculated as the ratio of the wet weight to the dry weight. The test was carried out in triplicate.

#### 2.4.5. Water Solubility

This was determined by immersing the films in distilled water at a film:water ratio of 1:10 for 48 h. The samples were exposed to a natural convection oven for 24 h at 60 °C to remove free water and were placed in a desiccator with P_2_O_5_ at 25 °C for 2 weeks to remove the tightly bound water. The solubility of the films was calculated from the initial and final weights. The test was performed in triplicate.

#### 2.4.6. Water Absorption Capacity

The test was carried out according to ASTM-D570 standard, using 25 mm by 60 mm specimens, which were placed in an oven at a temperature of 30 °C for 24 h to allow the samples to dry. After this time, they were immersed in distilled water for 2 h and weighed again to determine the amount of water absorbed.

#### 2.4.7. Water Vapor Permeability (WVP)

This was determined according to the protocol reported by Ortega-Toro et al. [31], following the ASTM E96-95 standard method (ASTM, 1995) with some modifications; a relative humidity gradient of 53% to 100% at a temperature of 25 °C was used. The films were selected for WVP testing based on the lack of physical defects. Distilled water was placed in Payne permeability cups to expose the film to 100% RH on one side. Once the films were secured, each cup was placed in a relative humidity-balanced cabinet at 25 °C. The RH of the cabinets (53%) was constant, using supersaturated solutions of magnesium nitrate-6-hydrate. The free film surface during film formation was exposed to the lowest relative humidity to simulate the actual application of the films on high water activity products when stored at intermediate relative humidity. The glasses were periodically weighed (0.0001 g) and the water vapor transmission (WVTR) was determined from the slope obtained from the regression analysis of the weight loss versus time data, once a steady state is reached, divided by the film area.

#### 2.4.8. Contact Angle

For the measurement of the contact angle, a space with a background and lighting suitable for taking photographs with light contrast was provided. Fruit peel or rind slices of 1 cm^2^ were placed on a flat level surface. A 0.1 mL drop was dropped onto the surface and photographs were taken at 10 s, 30 s and 60 s. Image analysis was then carried out using Adobe Photoshop to determine the contact angle formed between the emulsion drop and the analyzed surface. The greater the contact angle, the greater the wettability of the emulsion on the surface.

#### 2.4.9. Thermal Properties

The thermal properties were studied by thermogravimetric analysis (TGA) under nitrogen atmosphere in a Thermobalance TG-STDA Mettler Toledo model TGA/STDA851e/LF/1600 analyzer. TGA curves were obtained after conditioning the samples in the sensor for 5 min at 30 °C. The samples were then heated from 50 °C to 600 °C at a heating rate of 10 °C/min [10,34,35]. The first derivatives of thermogravimetry (DTG) curves, expressing the weight loss rate as the function of time, were also obtained using TA analysis software. All tests were carried out in triplicate.

#### 2.4.10. Antioxidant Activity

The antioxidant capacity of the films was determined using the 2,2-diphenyl-1-pikryl-hydroxyl (DPPH) reduction method [14]. 30 µL of samples diluted in water (1:10 for powder films) were mixed with 1 mL of 0.1 mM DPPH in methanol. The mixture was vortexed and allowed to stand at room temperature in the dark (40 min) before measuring absorbance at 517 nm. In the same way, 30 μL of samples were diluted in water (1:10 for the powder films) and 1 mL of ammonium solution of 2.2′-azino-bis-(3-ethylbenzothiazoline-6-sulfonic acid) (ABTS+) from Sigma Aldrich^®^ (St. Louis, MI, USA); the solutions were diluted in mixed methanol. After six minutes of reaction in the dark, the absorbance at 734 nm was monitored using a spectrophotometer (UV Visible Thermo Scientific Genesys 10S, Dreieich, Germany). The results were expressed in the IC_50_ parameter, which allows measuring the DPPH and ABTS+ radical scavenging capacity. The lower the IC50 value, the greater the antioxidant power it will have in the analyzed sample.

### 2.5. Application of Active Films on Baby Carrots

The application of active films based on starch and wheat gluten and containing essential oils of cinnamon and turmeric was carried out on baby carrots. Batches of 5 samples were evaluated for each coating in the study, and a negative blank without coating was also considered. Samples were stored at 25 °C and 75% RH for two weeks and monitored for weight loss, appearance, and fungal growth.

### 2.6. Statistical Analysis

Data for each test were analyzed statistically. Analysis of variance (ANOVA) was used to assess significance in the difference between factors and levels. Averages were evaluated using Fisher’s Least Significant Difference (LSD) test with 95% confidence. Data were analyzed using Statgraphics Plus for Windows 5.1 software (Manugistics Corp., Rockville, MD, USA).

## 3. Results

### 3.1. Active Films Characterization

#### 3.1.1. Thickness

Thickness is an important indicator of the physical properties of films, which is related to the volume of the film-forming liquid or the size of the spreading area [36]. Table 2 shows the thickness values for each of the formulations, where it is possible to observe the difference in the behavior when adding the Eos to the starch and gluten polymeric matrices. In the case of the starch matrix, the cinnamon and turmeric Eos reduced the thickness of the films by 3.9% and 19.6%, respectively; the opposite happened with the gluten films, where the incorporation of the cinnamon and turmeric EOs increased the thickness by 9.9% and 21.9%, respectively.

This behavior is due to the fact that when the EOs were incorporated into the starch films, they had a plasticizing effect on the matrix; Zhou et al. (2021) reported in the elaboration of edible films based on cassava starch that by incorporating cinnamon EO, the thickness of the films increased significantly between 73.00 and 137.20 μm with the increase in cinnamon EO content on the films [37]; this due to the formation of hydrophobic cinnamon EO microdroplets during the homogenization of the film forming solution, resulting in an increase in the thickness of the films [38].

In the gluten films, upon the addition of EOs, the free volume of the molecules increased, increasing the space between the chains of the gluten polymeric network; this thickness increase behavior was similar to that reported by Dong et al. (2022), in that by incorporating proteins, polysaccharides, and organic acids into wheat gluten films, their thickness increased. In particular, incorporating tartaric acid significantly increased the thickness, because tartaric acid could interact with the hydroxyl groups of the protein molecules, promoting an esterification/transesterification exchange reaction that interlaced the polymeric chains, creating a polymeric chain structure and thus increasing the thickness of the films [39,40].

#### 3.1.2. Morphology

The surface (Figure 1) and surface relief (Figure 2) of all the film formulations were performed by optical microscopy, with a 40× lens, in order to observe the dispersion and interfacial adhesion of the polymer system, since these characteristics play an important role in the other properties of the films. The adhesion of polymer systems could be also investigated through the measure of the minimum surface energy (Owens, D. K. (1970)) [41]. This energy is characterized by three main components (e.g., dispersive and polar), which can shed some light on chemistry surface and its wettability.

In general, all wheat gluten formulations with and without the addition of the EOs showed a more compact and uniform surface compared to the starch-based formulations. It can also be seen that with the incorporation of OEs, a change occurs in the surface of both starch and gluten films, with greater smoothness being observed in their surface topography. The cinnamon EO was the one that gave the best result in improving the surface smoothness of the films, as can be seen in formulations F3 and F4.

#### 3.1.3. Color Parameters

The effects of cinnamon and turmeric EO on the color parameters of the films are shown in Table 2. The values of L* represent the level of light and dark: negative values of a* represent green tones and positive values of b* represent yellow tones; h* represents the value of the total color difference; C* represents the value of the color intensity; and ΔE represents the value of the total color difference.

In general, the starch formulations modified with the EOs had a significant effect on ΔE (Table 2), with turmeric oil having the greatest effect on color changes. In addition, cinnamon oil significantly increased the total color difference value and decreased the color intensity value. Similar trends were observed with the addition of cinnamon EO in sago starch-based films [42]. This is due to the natural transparent color of the starch films, which facilitates the color change of the films, and the fact that the gluten films will not show a significant change in color, probably due to the natural yellowish color of the gluten films and the yellowish hue of EOs.

#### 3.1.4. Moisture, Water Solubility and Water Absorption Capacity

The interaction of films with water is an important characteristic of a film-forming material. The integrity and water resistance of films become very important when film materials are used as food packaging. The moisture content, water solubility and water absorption capacity are presented in Table 3. The starch-based films slightly decreased the moisture content by incorporating cinnamon and turmeric EOs, in turn significantly reducing the water solubility; however, the water absorption capacity increased. Amaral et al. (2019) observed that the addition of orange EO in starch films increased the moisture content; this is related to film breakdown, because the formation of porous structures in the oil films facilitates the insertion of water molecules between the polymer chains [43].

However, the results showed that the addition of cinnamon and turmeric EOs could improve the water resistance of the films. These results were probably attributed to the decrease in hydrophilicity of the matrix caused by the addition of the hydrophobic EO. Furthermore, the interaction between the EO components and the hydroxyl groups of the films reduced the interaction of the hydroxyl groups with the water molecules and led to a lower solubility of the films; however, due to the plasticizing effect that EOs may have on the matrix, the hydroxyl groups penetrate the starch polymer chains, increasing the space between the starch chains. Hence, there is more free space between the polymer chains which can be occupied by the water molecules, as illustrated in Figure 3.

In relation to the gluten matrix, the application of cinnamon EO represents a greater unfavourability because it increases the moisture content, water solubility, and water absorption capacity, the last two parameters being the highest of all formulations.

#### 3.1.5. Water Vapor Permeability

Water vapor permeability is an important characteristic of materials for food packaging purposes. Table 4 shows the water vapor permeability values of the active films. In general, the incorporation of the EOs into the starch-based matrix decreased the affinity of the matrix for water vapor, because the barrier formed by the EOs decreases the water vapor permeability. Turmeric oil showed the best results, decreasing this parameter from 1.6 to 1.29 g·mm/kPa·h·m^2^. With respect to gluten-based matrices, the increase in water vapor transmission may be due to the negative impact of cinnamon and turmeric EO on the microstructure of the films, as well as the generation of micropores and holes in the film structure, resulting in the diffusion of water vapor molecules. This may be attributed to the discontinuities caused in the polymer network by the lipid droplets of the EOs, which reduces the cohesion of the films. A similar trend has been found in fish gelatin and chitosan films containing EO of *Oregano vulgare* L. [44] and gelatin films incorporated with EO of Zataria multiflora [45].

#### 3.1.6. Contact Angle

The hydrophilicity of the film surface can be determined by the contact angle. Table 4 shows the contact angle of the active films at 10 and 30 s. The higher the contact angle, the higher the hydrophobicity of the films. Generally, when the contact angle is less than 65°, the film is hydrophilic (Oymaci and Altinkaya, 2016) [46]. In this work, the contact angle was taken at 10 and 30 s in all formulations, resulting in all films having a contact angle of less than 65° in both measurements, with no significant change in the contact angle. Dong et al. (2022) reported similar values in the contact angle in wheat gluten films with a value of 36.9°, although presenting low values in the angle in general the gluten-based films presented lower resistance to water, limiting their application [39].

#### 3.1.7. Thermal Properties

Figure 4 shows the TGA curves of the F1 (starch) and F2 (wheat gluten) films to ascertain their thermal stability. The starch and wheat gluten films showed multiple weight loss steps in the TG and DTG curves. The first step, occurring below ~200 °C with a weight loss of ~15%, is mainly attributed to the evaporation of free and bound water for both starch and gluten film [47,48,49].

Regarding the second step, with respect to wheat starch film (F1) in the range of ~200 °C to ~342 °C, the weight loss at this stage is related to the decomposition of low-weight molecules present in the matrix polymers, volatilization of glycerol, and starch degradation [50], with starch exhibiting a maximum thermal degradation temperature at 324 °C. The next stage can be attributed to the degradation of the film matrix in the range of ~342 °C to ~600 °C. Zhou et al. (2021) reported a value of 316 °C as the maximum thermal degradation temperature of cassava starch-based edible films [37].

Regarding wheat gluten film, the second phase of thermal degradation is divided into two parts: the first, in the range of ~200 to ~290 °C, is mainly due to the evaporation of glycerol; the second part, in the range of ~290 to ~340 °C, is due to the breaking of covalent peptide bonds in amino acid residues, 317 °C being the maximum thermal degradation temperature presented by gluten; and finally, above 340 °C, degradation is due to the cleavage of the S-S, O-N and O-O bonds of the protein molecules, resulting in the breakdown of the gluten matrix. Studies have reported that gluten-based films present a maximum thermal degradation temperature at 314 °C [51,52,53].

In general, the starch film presented higher thermal stability with respect to the gluten film, in terms of both the first phase of thermal degradation and in presenting a higher maximum temperature of thermal degradation.

### 3.2. Active Properties

#### Antioxidant Activity

Antioxidant films are a trend in the food industry for of the development of active food packaging. DPPH radicals are stable free radicals, and have been widely accepted as a tool to assess the radical scavenging activity of antioxidants. Table 5 gathers the antioxidant activity values expressed as DPPH radical scavenging activity and ABTS radical scavenging activity (IC_50_ values, the lower this parameter, the higher the antioxidant activity of the substance or material) of cinnamon, turmeric EOs, and starch and wheat gluten films containing EOs.

The differences in the antioxidant activity of different EOs were mainly due to differences in the types and amounts of antioxidant components present in the EOs; the main components of EOs are terpenes (monoterpenes and sesquerpenes), aromatic compounds (aldehydes, alcohols, phenol, methoxyderivatives, etc.), and terpenoids (isoprenoids) [48]. Turmeric EO exhibited higher antioxidant activity than cinnamon EO, with a greater effect on antioxidant activity noted in films containing turmeric EO. Consequently, the DPPH scan of the films containing the oil was significantly higher. The antioxidant activity of turmeric oil is attributed to its main components, such as α-Curcumin, α-Phellandrone, β-Bisabol, and Turmerone.

Cinnamon EO showed a lower antioxidant capacity compared to turmeric oil. This is attributed to the different components of cinnamon oil, such as Linalool, Menthol, Cinnamaldehyde and Eugenol, which react with the film microstructure, giving the film enhanced free radical scavenging capacity. The antioxidant activity can be attributed to the size of the oil droplets in the emulsion, because the smaller the droplet, the larger the area that coats the oil in the film. As such, the addition of EOs to the polymer matrix increases the antioxidant capacity of the final material. All films containing EOs showed antioxidant capacity, demonstrating that they can be applied in active packaging to extend the shelf life of food products. In particular, the cinnamon EO is used as an antimicrobial and antioxidant agent in food preservation, and is mainly composed of several active components, such as eugenol, cinnamic aldehyde, and cinnamic acid. In addition, it is considered to have anti-inflammatory, anti-stress, antifungal, anesthetic, and anti-allergic properties [53]. On the other hand, studies affirm that turmeric EO (TEO) has antioxidant, antibacterial, antiviral, anti-inflammatory, anticancer, and antimicrobial effects, and has been used as a preservative in food technology [54].

### 3.3. Coating Application on Baby Carrots

Figure 5 shows the weight loss of baby carrots with and without coating, stored at 25 °C and 75% RH for two weeks. The weight loss of the coated baby carrots is notably less accelerated than the uncoated baby carrots, the latter showing a constant weight loss during the two weeks of storage, while the former show a slower weight loss as time goes by. F5 was the coating that showed the best results, managing to reduce weight loss with respect to uncoated carrots by 55%, changing from losing 12.5 to 6.95 g of water/100 g of carrot in the two weeks of storage. Furthermore, it is evident that the incorporation of EOs in the formulations showed better results than the treatments without EOs (F1 and F2) in delaying the weight loss of the carrots, where this weight loss is directly related to the moisture requirement of these vegetable products. Therefore, there is a relationship with water vapor permeability, and as shown in Table 6, F5 had the lowest water vapor permeability of all treatments. However, despite the fact that F6 presented a higher value in water vapor permeability, it was one of the coatings that delayed weight loss the most; this behavior is possibly due to the fact that the turmeric EO presented good results in delaying the deterioration of the baby carrots, as shown in Table 6.

Table 6 shows the evolution of the properties of baby carrots with and without coating, stored at 75% RH and 25 °C for two weeks. The uncoated baby carrots presented a high value in their deterioration, where at the ninth day, all of them showed softening and 80% showed fungus, while the coated baby carrots showed a slower deterioration, where at the ninth day 40% of the coated baby carrots without EO showed a good appearance. EOs were shown to help prevent baby carrot deterioration, where on the ninth day, the 80% of the baby carrots coated with coatings containing EO showed good appearance, and the F6 formulation on that day prevented the presence of fungus. At the end of the two weeks of storage, the coatings containing EO performed better than the coatings without EO, with the turmeric EO the obtaining the best result in delaying the deterioration of the baby carrots. In one study, the authors evaluated the effect and shelf life of antioxidant-enriched edible coating on minimally processed carrots. They analyzed sliced carrots’ physicochemical, antioxidant, carotenoid, firmness, and microbial properties [54]. Likewise, in a study they evaluated the quality and shelf life of carrots incorporating an antimicrobial nanoemulsion (composed of citrus extract, cranberry juice, and EOs) as an active edible coating, and determined weight loss, texture, color, and its microbial quality [54].

## 4. Conclusions

Biodegradable active films based on starch and wheat gluten and containing cinnamon and turmeric Eos were obtained by casting method. The results indicated that the starch-based films showed a slight decrease in moisture content with the addition of Eos (up to 13.29%), but at the same time showed a significant reduction in water solubility (up to 28.4%). Gluten-based films did not present significant differences in these parameters, although the solubility in water tended to increase (up to 13.15%) with the addition of EOs. In general, the films presented good thermal stability and antioxidant capacity. In the carrot coating test, a decrease in weight loss of up to 44.44% and 43.33% was observed for the coatings based on starch and gluten, respectively, with the addition of turmeric EO. According to the DPPH and ABTS tests, turmeric oil has a higher antioxidant capacity than cinnamon oil, reflected in the antioxidant capacity of the films obtained. In addition, the starch matrix showed lower IC_50_ values than the gluten matrices, suggesting a higher retention of antioxidant components. According to the above, further research should be carried out on the development of biodegradable materials for food packaging, with the incorporation of promising essential oils such as turmeric oil, applying them to various food matrices to verify their effect on their stability during storage.

## Figures and Tables

**Figure 1 polymers-14-05077-f001:**
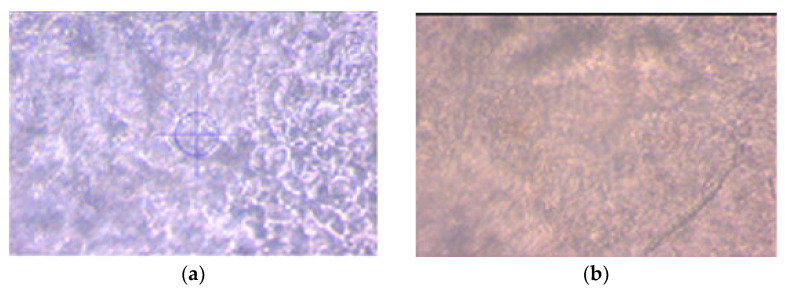
Optical micrographs of the films surface at 40× of films: (**a**) F1, (**b**) F2, (**c**) F3, (**d**) F4, (**e**) F5 and (**f**) F6.

**Figure 2 polymers-14-05077-f002:**
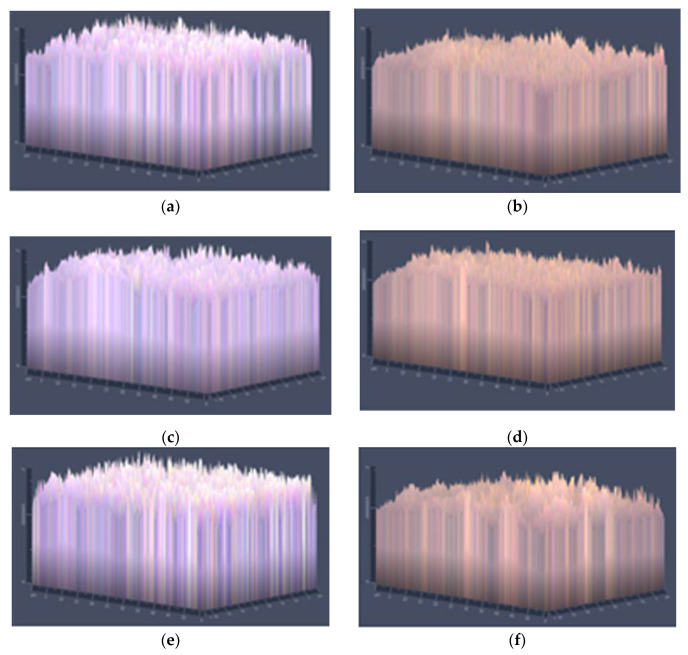
Optical micrographs of the films surface relief at 40× of films: (**a**) F1, (**b**) F2, (**c**) F3, (**d**) F4, (**e**) F5 and (**f**) F6.

**Figure 3 polymers-14-05077-f003:**
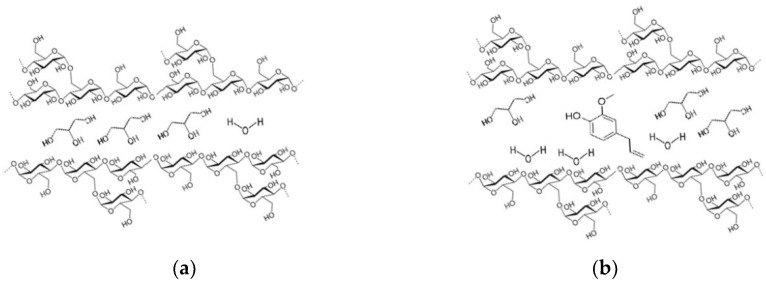
Representation of (**a**) molecules of starch plasticized with glycerol and water, and (**b**) molecules of starch, among which there are components of EOs that increase the intermolecular space and allow greater passage of water molecules, plasticizing the polymeric matrix.

**Figure 4 polymers-14-05077-f004:**
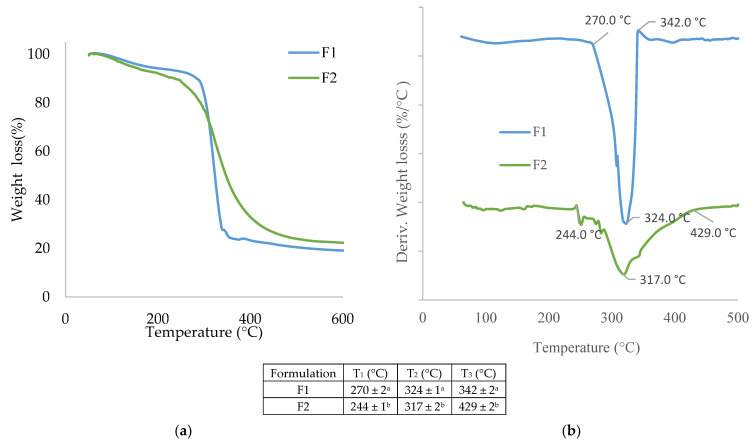
(**a**) Thermogravimetric analysis (TGA) and (**b**) first derivative (DTG) curves of films based on starch (F1) and wheat gluten (F2). Different superscripts within the same column indicate significant differences among samples (*p* < 0.05).

**Figure 5 polymers-14-05077-f005:**
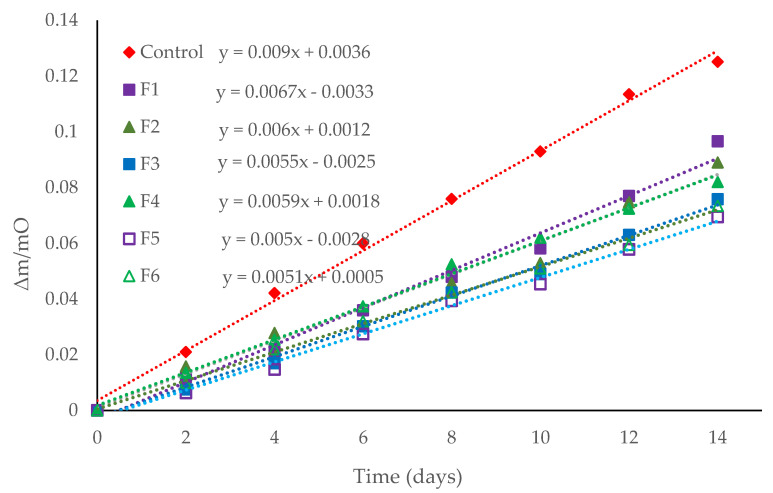
Weight loss of baby carrots with and without coating (F1, F2, F3, F4, F5, and F6), stored under 85% RH at 25 °C for two weeks.

**Table 1 polymers-14-05077-t001:** Experimental design expressed in mass fractions.

Films	Starch	Wheat Gluten	Glycerol	Cinnamon Essential Oil (EO_CN_)	Turmeric Essential Oil (EO_TU_)
F1	0.800	0.000	0.200	0.000	0.000
F2	0.000	0.800	0.200	0.000	0.000
F3	0.782	0.000	0.195	0.023	0.000
F4	0.000	0.782	0.195	0.023	0.000
F5	0.782	0.000	0.195	0.000	0.023
F6	0.000	0.782	0.195	0.000	0.023

**Table 2 polymers-14-05077-t002:** Means and deviation standard of color parameters of the films.

Film	Thickness (µm)	L *	a *	b *	h *	C *	ΔE
F1	102 ± 11 ^b^	91.3 ± 1.2 ^c^	−1.16 ± 0.03 ^d^	5.62 ± 0.12 ^c^	101.7 ± 0.5 ^c^	5.7 ± 0.2 ^c^	---
F2	91 ± 13 ^a^	38.1 ± 0.3 ^b^	−0.67 ± 0.02 ^c^	6.79 ± 0.08 ^e^	95.6 ± 0.6 ^b^	6.8 ± 0.2 ^d^	---
F3	98 ± 9 ^ab^	36.1 ± 0.2 ^b^	−0.48 ± 0.03 ^b^	−0.05 ± 0.02 ^a^	264 ± 2 ^d^	0.48 ± 0.05 ^a^	55.49 ± 0.5 ^c^*
F4	100 ± 16 ^b^	37.8 ± 0.3 ^b^	−0.47 ± 0.04 ^b^	6.16 ± 0.03 ^d^	94.4 ± 0.5 ^b^	6.2 ± 0.9 ^cd^	0.70 ± 0.02 ^a^**
F5	82 ± 14 ^a^	31.6 ± 0.3 ^a^	−0.27 ± 0.02 ^a^	0.61 ± 0.03 ^b^	114.2 ± 0.6 ^d^	0.66 ± 0.02 ^b^	59.928 ± 0.3 ^d^*
F6	111 ± 12 ^b^	37.2 ± 0.2 ^b^	−0.21 ± 0.02 ^a^	8.72 ± 0.04 ^f^	91.3 ± 0.4 ^a^	8.72 ± 0.05 ^e^	2.15 ± 0.05 ^b^**

a–f: Different superscripts within the same column indicate significant differences among samples (*p* < 0.05). * Parameter compared to F1. ** Parameter compared to F2.

**Table 3 polymers-14-05077-t003:** Moisture content (MC), water solubility (WS), and water absorption capacity (WAC) of the films.

Film	MC (%)	WS (%)	WAC (%)
F1	11.58 ± 1.17 ^b^	45.7±12.8 ^b^	0.86 ± 0.03 ^a^
F2	11.21 ± 0.22 ^b^	46.4 ± 9.4 ^b^	1.07 ± 0.54 ^b^
F3	10.81 ± 0.51 ^ab^	38.1 ± 4.9 ^a^	1.24 ± 0.52 ^b^
F4	11.32 ± 0.28 ^b^	52.5 ± 1.6 ^b^	1.36 ± 0.06 ^b^
F5	10.04 ± 0.27 ^a^	32.7 ± 3.1 ^a^	1.22 ± 0.13 ^b^
F6	10.74 ± 0.62 ^ab^	48.4 ± 4.3 ^b^	0.91 ± 0.14 ^a^

a,b: Different superscripts within the same column indicate significant differences among samples (*p* < 0.05).

**Table 4 polymers-14-05077-t004:** Water vapor permeability (WVP) and contact angle (CA) at 10 and 30 s of the films.

Film	WVP (g·mm/kPa·h·m^2^)	CA-10s (°)	CA-30s (°)
F1	1.60 ± 0.10 ^b^	35.9 ± 0.7 ^a^	36.8 ± 0.7 ^a^
F2	1.44 ± 0.09 ^a^	38.70 ± 1.01 ^b^	39.4 ± 0.8 ^b^
F3	1.53 ± 0.09 ^ab^	36.8 ± 0.3 ^a^	37.6 ± 0.3 ^a^
F4	1.57 ± 0.09 ^b^	39.1 ± 0.2 ^b^	39.9 ± 0.3 ^b^
F5	1.29 ± 0.08 ^a^	36.9 ± 0.4 ^a^	37.8 ± 0.3 ^a^
F6	1.75 ± 0.11 ^b^	38.9 ± 0.5 ^b^	39.9 ± 0.3 ^b^

a,b: Different superscripts within the same column indicate significant differences among samples (*p* < 0.05).

**Table 5 polymers-14-05077-t005:** Antioxidant activity of cinnamon (EO_CN_) and turmeric EOs (EO_TU_).

Sample	DPPH IC_50_ (EO mL/DPPH mg)	ABTS IC_50_ (EO mL/DPPH mg)
EO_CN_	146.6 ± 1.5 ^a^	28.4 ± 0.5 ^a^
EO_TU_	14.2 ± 0.8 ^b^	2.62 ± 0.13 ^b^
	**DPPH IC_50_ (film mg/DPPH mg)**	**ABTS IC_50_ (film mg/DPPH mg)**
F3	191.4 ± 1.3 ^c^	53.2 ± 0.8 ^c^
F4	213 ± 1 ^d^	67.6 ± 0.9 ^d^
F5	45.6 ± 0.9 ^a^	28.6 ± 0.5 ^a^
F6	60.2 ± 0.8 ^b^	38.4 ± 0.5 ^b^

a–d: Different superscripts within the same column indicate significant differences among samples (*p* < 0.05).

**Table 6 polymers-14-05077-t006:** Physical properties of baby carrots with and without coating (F1, F2, F3, F4, F5, and F6), stored under 85% RH at 25 °C for two weeks.

Day	Films	Visual Aspect (%)	Softening (%)	Presence of Fungi
0	Control	100	0	0
	F1	100	0	0
	F2	100	0	0
	F3	100	0	0
	F4	100	0	0
	F5	100	0	0
	F6	100	0	0
3	Control	60	40	0
	F1	60	40	0
	F2	60	40	0
	F3	80	20	0
	F4	80	20	0
	F5	100	0	0
	F6	100	0	0
6	Control	20	80	40
	F1	60	40	40
	F2	60	40	20
	F3	80	20	0
	F4	80	20	0
	F5	80	20	0
	F6	100	0	0
9	Control	0	100	80
	F1	40	60	60
	F2	40	60	60
	F3	80	20	20
	F4	80	20	20
	F5	80	20	20
	F6	80	20	0
12	Control	0	100	100
	F1	20	80	60
	F2	20	80	80
	F3	60	40	20
	F4	60	40	40
	F5	60	40	20
	F6	60	40	20
15	Control	0	100	100
	F1	0	100	80
	F2	20	80	80
	F3	40	60	40
	F4	40	60	40
	F5	60	40	40
	F6	60	40	20

## Data Availability

Not applicable.

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
