# Peer review of "Active Films Based on Starch and Wheat Gluten (Triticum vulgare) for Shelf-Life Extension of Carrots"

_polymers, 2022, doi:10.3390/polym14235077_

Round 1

Reviewer 1 Report

Revision for Polymers (ISSN 2073-436)

Manuscript ID: polymers-2020811

Title: Active films Based on Starch and Wheat Gluten (Triticum Vulgare) for Shelf-Life Extension of Plant Products

List of authors: Andrés Felipe Rivera Leiva , Joaquín Hernández- Fernández * , Rodrigo Ortega Toro *

This research paper deals with the preparation of active films based on starch and wheat gluten containing cinnamon and turmeric essential oils by solvent casting method. Also, the authors investigated several characteristics (i.e. morphology, optical, thermal, antioxidant and barrier properties) of different film formulations, including the active properties on baby carrots regarding weight loss, appearance and fungal growth. I consider the research work a valuable activity and it can open many future prospective in the field of biodegradable biopolymers with the incorporation of active ingredients. As the subject is interesting, I am willing to recommend minor revisions with pending manuscript decision. I will gladly be able to review the modified manuscript once the following points have been fully addressed: 

1) Page 1, lines 16-28 - I would suggest to add some quantitative results in the abstract to highlight the goals achieved in the research.

2) Page 1, lines 41-43 - I would suggest to add more references to support this statement.

3) Page 2, lines 52-53 - Could you provide some previous results reported in the literature regarding the use of starch-based films?

4) Page 2, line 67 - There are some missing references at this point.

5) Page 2, lines 80-81 - Could you report a list of such methods with related reference?

6) Page 2, lines 91-96 - Could you report in the list the characterization techniques used with their acronyms?

7) Page 3, lines 111-116 - Could you add some photos to make an easier understanding of each step for the reader?

8) Page 4, lines 150-153 - More details should be provided regarding the optical microscope used, as for example city and country where the manufacturer is based.

9) Page 4, lines 156-159 - Could you add a reference for such equation (1)?

10) Page 6, lines 261-269 - I would suggest to perform a dynamic mechanical analysis on the gluten polymeric network to characterize the viscoelastic behavior of such material. For example, the evaluation of tanδ and elastic modulus versus temperature would be interesting.

11) Page 6, lines 272-275 - The adhesion of polymer systems could be also investigated through the measure of the minimum surface energy (Owens, D. K. (1970). Some thermodynamic aspects of polymer adhesion. Journal of applied polymer science, 14(7), 1725-1730). This energy is characterized by three main components (e.g. dispersive and polar), which can shed some light on chemistry surface and its wettability.

12) Page 6, lines 278-281 - I would suggest to avoid the repetition of "morphology" and some descriptions of main observations should be provided.

13) Page 9, lines 316-325 - I would suggest to add a small scheme to provide a visual representation of the plasticizing effect.

14) Page 14, lines 471-493 - The conclusion could be reduced to highlight the main results and achievements. I would also suggest to add some quantitative results.

Author Response

Reviewer 1

Comment: This research paper deals with the preparation of active films based on starch and wheat gluten containing cinnamon and turmeric essential oils by solvent casting method. Also, the authors investigated several characteristics (i.e. morphology, optical, thermal, antioxidant and barrier properties) of different film formulations, including the active properties on baby carrots regarding weight loss, appearance and fungal growth. I consider the research work a valuable activity and it can open many future prospective in the field of biodegradable biopolymers with the incorporation of active ingredients. As the subject is interesting, I am willing to recommend minor revisions with pending manuscript decision. I will gladly be able to review the modified manuscript once the following points have been fully addressed: 

Answer: Thanks for the comments, they helped us to improve the manuscript

Comment: 1) Page 1, lines 16-28 - I would suggest to add some quantitative results in the abstract to highlight the goals achieved in the research.

Answer: It was done

Comment: 3) Page 2, lines 91-96 - Could you report in the list the characterization techniques used with their acronyms?

Answer: It was done

Comment: 4) Page 3, lines 111-116 - Could you add some photos to make an easier understanding of each step for the reader?

Answer: Unfortunately, we do not have photos of this procedure. The student in charge of the laboratory tests did not take photos of this part of the process. However, it is a simple procedure, it take a portion of wheat flour and subject it to repeated washing with plenty of water and constant kneading; in this way, the starch separates in the water , and the gluten forms a rubbery amorphous mass that it is not soluble in water. In this manner, the two components are separated.

Comment: 5) Page 4, lines 150-153 - More details should be provided regarding the optical microscope used, as for example city and country where the manufacturer is based.

Answer: It was added

Comment: 6) Page 6, lines 261-269 - I would suggest to perform a dynamic mechanical analysis on the gluten polymeric network to characterize the viscoelastic behavior of such material. For example, the evaluation of tanδ and elastic modulus versus temperature would be interesting.

Answer: The comment is appreciated. It would be interesting to have such additional data. However, the study of the viscoelastic behaviour of the material has yet to be objective for this phase of the investigation. We are preparing new contributions where we will gladly consider the suggestion, and this type of analysis will be carried out.

Comment: 7) Page 6, lines 272-275 - The adhesion of polymer systems could be also investigated through the measure of the minimum surface energy (Owens, D. K. (1970). Some thermodynamic aspects of polymer adhesion. Journal of applied polymer science, 14(7), 1725-1730). This energy is characterized by three main components (e.g. dispersive and polar), which can shed some light on chemistry surface and its wettability.

Answer: The phrase is appreciated and was added to the manuscript, as well as the respective reference.

Comment: 8) Page 6, lines 278-281 - I would suggest to avoid the repetition of "morphology" and some descriptions of main observations should be provided.

Answer: It was done

Comment: 9) Page 9, lines 316-325 - I would suggest to add a small scheme to provide a visual representation of the plasticizing effect.

Answer: Figure 3 was added with the small scheme about plasticizing effect

Comment: 10) Page 14, lines 471-493 - The conclusion could be reduced to highlight the main results and achievements. I would also suggest to add some quantitative results.

Answer: conclusion was improved

Reviewer 2 Report

The manuscript “Active films Based on Starch and Wheat Gluten (Triticum Vulgare) for Shelf-Life Extension of Plant Products” presents good results, and the proposed biodegradable active biofilms have the potential to be applied to several food products.

Some minor corrections:

line 33-34 - This section needs reference:  https://doi.org/10.3390/molecules27175604.

line 38 – 40 - This section needs additional references: https://doi.org/10.3390/antiox11091729, https://doi.org/10.1016/j.tibtech.2019.04.011

Based on this last reference, an important aspect indicated in the present article is the term “biodegradable” – but the authors did not mention any biodegradability analysis which is an important aspect.

line 43 – 45 - This section needs a reference, i.e. https://doi.org/10.3390/gels8080524

line 53 – please finish the sentence

line 67 – please correct the error

line 73 – essential oil has already been abbreviated at line 59 – please use the abbreviated form afterwards – revise the whole manuscript (i.e. line 77, 92, etc)

line 83 – the “solvent casting” method needs a reference i.e. https://doi.org/10.1016/j.carbpol.2007.05.041

line 123 – please insert a space between the number and degree sign – revise the whole manuscript

line 151 – please specify the type and producer of the used optical microscope – the same for the desiccator (line 172)

line 213 - regarding the number of performed tests, it is sufficient to be added to the statistical analyses.

lines 292 – 296 – please rephrase this sentence; it is too long, and hard to understand

Some future perspectives could also be inserted in the conclusion section.

Author Response

Reviewer 2

Comment: The manuscript “Active films Based on Starch and Wheat Gluten (Triticum Vulgare) for Shelf-Life Extension of Plant Products” presents good results, and the proposed biodegradable active biofilms have the potential to be applied to several food products.

 Answer: Thanks for the comments, they helped us to improve the manuscript

Comment: line 33-34 - This section needs reference:  https://doi.org/10.3390/molecules27175604.

Answer: It was done

Comment: line 38 – 40 - This section needs additional references: https://doi.org/10.3390/antiox11091729, https://doi.org/10.1016/j.tibtech.2019.04.011

Answer: It was done

Comment: Based on this last reference, an important aspect indicated in the present article is the term “biodegradable” – but the authors did not mention any biodegradability analysis which is an important aspect.

Answer: The comment is appreciated. Biodegradation tests have not been carried out on these matrices since it is known from previous experiences and literature that starch and gluten matrices degrade under composting conditions. However, it will be taken into account for future research.

Comment: line 43 – 45 - This section needs a reference, i.e. https://doi.org/10.3390/gels8080524

Answer: It was done

Comment: line 53 – please finish the sentence

Answer: It was done

Comment: line 67 – please correct the error

Answer: the sentence was improved

Comment: line 73 – essential oil has already been abbreviated at line 59 – please use the abbreviated form afterwards – revise the whole manuscript (i.e. line 77, 92, etc)

Answer: It was done

Comment: line 83 – the “solvent casting” method needs a reference i.e. https://doi.org/10.1016/j.carbpol.2007.05.041

Answer: It was done

Comment: line 123 – please insert a space between the number and degree sign – revise the whole manuscript

Answer: It was done

Comment: line 213 - regarding the number of performed tests, it is sufficient to be added to the statistical analyses.

Answer: It was added as part of Figure 4

Comment: lines 292 – 296 – please rephrase this sentence; it is too long, and hard to understand

Answer: The sentence was improved

Comment: Some future perspectives could also be inserted in the conclusion section.

Answer: It was added

Round 2

Reviewer 2 Report

The authors considerably improved the manuscript, and included the requested future perspectives. As follows the manuscript can be accepted for publication.